# Low-Temperature SCR Catalyst Development and Industrial Applications in China

Hongtai Zhu [1,2,3], Liyun Song [1,2,*], Kai Li [3], Rui Wu [3], Wenge Qiu [1,2] and Hong He [1,2,3,*]

1   Key Laboratory of Beijing on Regional Air Pollution Control, Beijing University of Technology, Beijing 100124, China; zhuhongtai624@163.com (H.Z.); qiuwenge@bjut.edu.cn (W.Q.)
2   Beijing Key Laboratory for Green Catalysis and Separation, Beijing University of Technology, Beijing 100124, China
3   Advanced E-Catal. Corporation, Ltd., Beijing 100025, China; kkkk08052121@sina.com (K.L.); wuruicoco@sina.com (R.W.)
*   Correspondence: songly@bjut.edu.cn (L.S.); hehong@bjut.edu.cn (H.H.)

**Abstract:** In recent years, low-temperature SCR (Selective Catalytic Reduction) denitrification technology has been popularized in non-power industries and has played an important role in the control of industrial flue gas $NO_x$ emissions in China. Currently, the most commonly used catalysts in industry are $V_2O_5$-$WO_3$($MoO_3$)/$TiO_2$, $MnO_2$-based catalysts, $CeO_2$-based catalysts, $MnO_2$-$CeO_2$ catalysts and zeolite SCR catalysts. The flue gas emitted during industrial combustion usually contains $SO_2$, moisture and alkali metals, which can affect the service life of SCR catalysts. This paper summarizes the mechanism of catalyst poisoning and aims to reduce the negative effect of $NH_4HSO_4$ on the activity of the SCR catalyst at low temperatures in industrial applications. It also presents the outstanding achievements of domestic companies in denitrification in the non-power industry in recent years. Much progress has been made in the research and application of low-temperature $NH_3$-SCR, and with the renewed demand for deeper $NO_x$ treatments, new technologies with lower energy consumption and more functions need to be developed.

**Keywords:** selective catalytic reduction; nitrogen oxides; low temperature; catalyst poisoning; non-power

## 1. Introduction

Nitrogen oxides ($NO_x$), including NO and $NO_2$, are considered as the main air pollutants from industrial and automobile exhausts, which have caused a lot of environmental problems, such as haze issues and ozone depletion. Until now, the selective catalytic reduction of $NO_x$ by ammonia ($NH_3$-SCR) is accepted to be an effective method to eliminate $NO_x$ pollutants [1].

In SCR denitration technology, the most important thing is the catalyst that should possess the high activity, excellent sulfur oxides and water resistance abilities. The traditional V-based catalysts showed good deNO$_x$ performance at 300–420 °C, which have been well used to abate the flue gas from the power plants. Due to the wide applications of SCR technology, the $NO_x$ emissions from power industry have been well controlled, while $NO_x$ emission control in the non-power industry faces severe challenges due to the low temperature of the flue gas, which is usually below 300 °C. Therefore, it is difficult to use traditional SCR catalysts for the gas pollutant treatment in the non-power industry. In the past ten years, Chinese scholars and industries have made great efforts on denitration in the non-power industry and made remarkable achievements. Wang and Dong et al. [1,2] present the detailed information concerning $NH_3$-SCR in some non-power industries, showing that the gas condition was more complex and fluctuated than that from the power plant. This review focus on the research and development in low-temperature SCR and its application in non-power industry, especially on the anti-sulfur and water resistance properties of the catalyst, as well as on how to reduce the negative impact of

ammonium bisulfate (ABS, $NH_4HSO_4$) on the catalyst activity at low temperatures in industrial applications.

## 2. Low-Temperature SCR Process

Usually, the flue gas temperature of the industrial process, such as in coking and steel sintering processes, is lower than 300 °C and contains many constituents with low concentrations, such as sulfur dioxide and alkali metal salts, and large amounts of water are also present. Thus, the SCR catalyst must be active in the low-temperature range (typically in the range of 160–300 °C) and stable under harsh gas conditions with good sulfur oxides and water resistance performance [3]. In the typical flue gas treatment system of power plants in China, which usually employ high-dust SCR system to control $NO_x$ emissions, the SCR reactor is located upstream of the particulate control devices and flue gas desulfurization (FGD) system. The so-called "medium- and high-temperature SCR" units can be operated in the temperature range of 280–400 °C. For $NO_x$ emission control in the non-power industry, especially in the coking and steel sintering industry, the "low-dust" or "tail-end" configuration of SCR technology should be adopted to decrease the impacts of $SO_2$ and dust on the SCR catalyst. The low-temperature SCR reactor is located downstream of the particulate control devices and flue gas desulfurization (FGD) system. The $SO_2$ concentration and dust amount in inlet of the SCR reactor should be lower than 35 and 5 mg/m$^3$, respectively, to meet the ultra-low emission standards. In the key areas of coal-fired boilers, the value should be lower than 50 and 20 mg/m$^3$. In this case, the operating temperature of the SCR unit can be decreased to 160 °C. Although $SO_2$ and dust in the flue gas have very low values, the SCR catalyst needs to operate for three years at a low temperature, posing a severe challenge to SCR catalyst technology.

## 3. SCR Catalysts

Low-temperature SCR technology is an economic and effective process in abating the $NO_x$ pollutants emitted from the non-power industry. Based on the consensus of the advantages that low-temperature SCR technology possesses, in the last decade, considerable research in China have been devoted to developing catalysts that can work well under low-temperature conditions. The development and research in SCR catalysts have been reviewed and summarized [2,3]. As reported, many kinds of low-temperature SCR catalyst system have been proposed and investigated. The main low-temperature SCR catalyst systems include $V_2O_5$-$WO_3$($MoO_3$)/$TiO_2$, Mn complex oxides, $CeO_2$-based and zeolite catalysts.

### 3.1. $V_2O_5$-$WO_3$($MoO_3$)/$TiO_2$

As a typical and efficient catalyst, a $V_2O_5$-$WO_3$($MoO_3$)/$TiO_2$ catalyst used in $NH_3$-SCR technology has been commercialized for several decades. The typical commercial catalyst used in power stations presents low activity at low temperatures (below 300 °C) and cannot meet the need to abate $NO_x$ from non-power industries.

In the past decades, various transition-metal oxides have been researched as catalysts for $NH_3$-SCR at low temperatures. In order to meet the need of activity, stability, and resistance of $SO_2$ and $H_2O$, plenty of methods have been tried to improve SCR performance.

The most direct and convenient method to improve the low temperature activity of $V_2O_5$-$WO_3$($MoO_3$)/$TiO_2$ catalyst is to appropriately increase the loading of $V_2O_5$. However, when $V_2O_5$ loading increases, the oxidative ability of the catalyst will be increased leading to the enhancement of $SO_2$ conversion. This is not allowed for the low-temperature SCR technology. Therefore, the catalyst needs to coordinate the redox activity and the surface acid property, reduce the adsorption of $SO_2$ on the catalyst surface and suppress the oxidation of $SO_2$. In another way, the $NH_3$ adsorbed on the Lewis acid sites ($V^{5+}$-O) on $V_2O_5$-$WO_3$($MoO_3$)/$TiO_2$ catalysts can react with NO at low temperatures [4]. By increasing the surface acidity and inhibiting the oxidative ability over the SCR catalysts, the operating temperature window of the $V_2O_5$-$MoO_3$/$TiO_2$ catalyst is expanded to the range of 160–400 °C, which also shows acceptable $SO_2$ and $H_2O$ resistance at low temperatures [5–7].

The $V_2O_5$-$MoO_3$/$TiO_2$ catalysts have been used intensively in denitration reaction projects of coking sintering, refuse incinerators and other non-electric industries in China.

Another way to improve the low-temperature SCR activity of the $V_2O_5$-$WO_3$($MoO_3$)/$TiO_2$ catalyst is through modification and doping by introducing other elements into the catalyst system, which is easy to achieve in practice due to the convenience in the preparation. For example, Zhang et al. [8] investigated the effect of Mn, Cu, Sb, and La doping on the SCR performance of the $V_2O_5$-$WO_3$($MoO_3$)/$TiO_2$ catalyst. The investigation showed that Mn and Cu could enhance the redox property and weak surface acidity, while Sb and La addition showed promotion in the amount of acid sites. Liang et al. [9] demonstrated that a 3% addition of $CeO_2$ improved the $NH_3$ adsorption performance, NO oxidation, and sulfur oxide and the water-resistance of the $V_2O_5$-$WO_3$/$TiO_2$ catalyst.

### 3.2. MnO_2-Based Catalysts

Manganese-containing catalysts have been paid enough attention due to their variable valence states and excellent redox properties. However, for its poor $N_2$ selectivity and easy deactivation by $SO_2$, the catalyst only containing manganese oxide is extremely restricted in the SCR process. Mn-based composite oxides are popular and have proven to be effective catalysts with an enhanced SCR performance [10].

Over $MnO_x$ catalyst, $NH_3$ species adsorbed on Lewis acid sites ($Mn^{3+}$) were active at low temperatures. Bidentate nitrates were inactive at low temperatures (below ~225 °C), but active at higher temperatures [11].

Mn-based composite oxides possess excellent redox properties due to their various valence state, which are a benefit to enhance the process of NO oxidation to $NO_2$. The formation of $NO_2$ from NO oxidation is considered as a key factor in low-temperature activities because a certain concentration of $NO_2$ gives an enhancement of the "Fast SCR" reaction at low temperatures. Chen et al. [12] proposed that the redox cycle between $Cr^{5+} + 2Mn^{3+} \leftrightarrow Cr^{3+} + 2Mn^{4+}$ promoted the oxidization of NO to $NO_2$ at low temperatures. Liu et al. [13] reported that an urchin-like $MnCrO_x$ catalyst possessed good $NH_3$-SCR activity in the temperature range of 150–350 °C and improved $SO_2$ resistance.

Gao et al. [14] discovered that $CoMnO_x$ showed high $NH_3$-SCR activity at low temperatures and delayed the trend of $SO_2$ poisoning. Zhao et al. [15] found that a lamellar $CoMnO_x$ composite oxide could provide more Lewis acid sites and surface oxygen species than those of $CoMnO_x$ nanoparticles. Wang et al. [16] reported that ballflower-like $CoMnO_x$ catalyst exhibited good SCR activity and $N_2$ selectivity in the temperature range of 150–350 °C, showing a certain amount of $SO_2$ resistance and durability.

The doped component was usually considered to give a promotion of surface lattice oxygen species. Sun et al. [17] investigated the $NH_3$-SCR activity over the Nb-doped $Mn/TiO_2$ catalysts with the optimum Nb/Mn molar ratio of 0.12. Rare earth elements were also used in the modifications. Liu et al. [18] and Xu et al. [19] investigated the effect of the introduction of Sm to $Mn$-$TiO_x$ catalysts. The introduced Sm could improve the dispersion of manganese oxide on the surface of the catalysts, resulting in increases in surface area, the amount of weak Lewis acid sites and surface oxygen.

Among these catalysts, spinel-type materials containing manganese attracted interest in SCR due to their special spatial structures and physical-chemical properties. Gao et al. [20] reported that a $Zr^{4+}$ cation doped $MnCr_2O_4$ spinel, the zirconium incorporated in the crystal of $MnCr_2O_4$ produced higher levels of beneficial $Mn^{3+}$, $Mn^{4+}$ and $Cr^{5+}$ species, and showed an increase in the acidity and redox ability.

However, these catalysts are very sensitive and exhibit unsatisfactory $N_2$ selectivity [21]. The stability in the presence of $SO_2$ and $H_2O$ in the flue gas is still a problem for $MnO_2$-based catalysts.

### 3.3. CeO_2-Based Catalysts

He et al. [22,23] reported that the crystal structure, crystallite size and catalytic $NH_3$-SCR activity over the $CeO_2$-based catalysts presented a regular change with the increase in

CeO$_2$ concentration. Particularly, the CeO$_2$-TiO$_2$ (1:1 in weight) catalyst with an amorphous structure showed a higher BET surface area and a stronger surface acidity than other samples. Meanwhile, favorable Ce$^{3+}$ and the surface-adsorbed oxygen benefited the adsorption of NO$_x$ and NH$_3$ molecules, which could enhance NH$_3$-SCR activity.

In the past years, tungsten or molybdenum addition in ceria-based catalysts was paid some attention. Jiang et al. [24] demonstrated that the introduction of WO$_3$ could improve SCR activity over the CeWTiO$_x$ catalysts due to the enhanced dispersion of Ce species over TiO$_2$ and the amount of Ce$^{3+}$ and chemisorbed oxygen. Li et al. [25] investigated the adsorption and reactivity of NH$_3$ and NO over the CeWTiO$_x$ catalyst, showing that over 90% of NO conversion can be obtained in the temperature range of 250–500 °C. Liu et al. [26] reported that the WO$_3$/TiO$_2$@CeO$_2$ core–shell catalyst present a synergistic effect of redox properties and acidity, which is in favor of the excellent NH$_3$-SCR activity and better SO$_2$ resistance. Kim et al. [27] found monomeric W in CeO$_2$/TiO$_2$ catalyst enhance the SCR reaction activity at low temperatures due to the increased NO adsorption and the formation of unstable NO$_x$ adsorption species. Dong et al. [28] presented that the coverage of MoO$_3$ weakened the adsorption of nitrate species over the CeO$_2$-TiO$_2$ catalyst, giving an increase in the number of Brønsted acid sites. For the CeO$_2$-based catalyst, cerium sulfate, which is formed in reaction with SO$_2$ in the flue gas during the SCR process at high temperatures, has attracted wide attention. Fan et al. [29] showed that NH$_3$-SCR reaction over CeO$_2$/TiO$_2$-ZrO$_2$-SO$_4^{2-}$ mainly followed the L–H mechanism at a low temperature (250 °C) and the E–R mechanism at 350 °C.

Generally, for the limitation of sulfate formation and low temperature SCR performance, CeO$_2$-based SCR catalyst does not seem an optimal choice for NO$_x$ elimination under low-temperature conditions at present.

### 3.4. MnO$_2$-CeO$_2$ Catalysts

Rare-earth metal oxides, such as Ce, have been frequently adopted to modify MnO$_x$ as an efficient low-temperature NH$_3$-SCR catalyst due to their incomplete 4f and empty 5d orbitals [30]. Leng et al. [31] synthesized MnCeTiO$_x$ catalysts and compared the NH$_3$-SCR activity over the samples with different Mn/Ce mol ratios in the low-temperature range. The results showed that the low-temperature SCR activity over MnCeTiO$_x$ compositions was greatly improved due to the incorporation of Mn, and the best performance (~100% NO conversion and above 90% N$_2$ yields) across the temperature range of 175–400 °C at GHSV of 80,000 h$^{-1}$.

In the published lectures, composite transition metal oxides usually showed a higher activity than single oxide materials. Zhang et al. [32] demonstrated the enhanced electron mobility effect that originated from MnO$_x$ and CeO$_x$, which enhanced low-temperature deNO$_x$ efficiency. Compared to the single composition of CeO$_2$, MnO$_x$ could increase the pore volume and pore diameter, and enhance the adsorption of NO and NH$_3$ as well as in the concentrations of Ce$^{3+}$ on the CeO$_2$-MnO$_x$ catalyst, which is beneficial to increase the redox properties [33].

Yang et al. [34] studied SCR activity over the activated carbon supported Mn-Ce oxide catalysts modified by Fe, on which ca. 90% NO conversion was obtained at 125 °C with GHSV of 12,000 h$^{-1}$. Zhu et al. [35] synthesized a 3D-structured MnOx-CeO$_2$/reduced graphene oxide, giving a NO$_x$ conversion of 99% at 220 °C with GHSV of 30,000 h$^{-1}$.

### 3.5. Zeolite SCR Catalysts

Ion-exchanged zeolite catalysts with small pores have been accepted as optimum SCR catalysts in NO$_x$ elimination from diesel engine exhaust. Among them, copper or iron exchanged zeolites with a chabazite (CHA) structure, such as Cu/SAPO-34 and Cu/SSZ-13, have received significant attention due to their excellent SCR performance, wide temperature window and thermal stability in harsh conditions [36]. Cu-SSZ-13 exhibits excellent SCR activity (>80% NO conversion) and N$_2$ selectivity in a wide temperature range of 150−450 °C [37]. Cu/SAPO-34 prepared by a hard-template method using CaCO$_3$ as tem-

plate present $NO_x$ convention above 90% at 170–480 °C, even introducing 10% $H_2O$ [38]. A heterobimetallic FeCu-SSZ-13 zeolite with high crystallinity was prepared by an economic and sustainable one-pot synthesis strategy, which presents a wide reaction temperature window, excellent hydrothermal stability, high $H_2O$ and $SO_2$ tolerance, and good gaseous hourly space velocity flexibility [39].

Zeolite catalysts for SCR has been developed rapidly these years, offering a great contribution to abate the $NO_x$ reduction. However, they may be not the optimum choice for $NO_x$ elimination in stationary sources due to the limitations of cost and synthesis strategy.

## 4. Deactivation of Low Temperature SCR Catalyst

The flue gas emitted from industrial combustion processes usually contains $SO_2$, moisture, alkali metals, among others. These factors can reduce the activity of catalysts. Figuring out the mechanism of poisoning and how to reduce the effect of these factors has been a problem faced by the researchers as there are many reports discussing this issue.

### 4.1. SO₂ Poisoning

A primary mechanism of $SO_2$ deactivation over the catalyst at low temperatures can be observed in three ways. (i) While $SO_2$ can be oxidized to $SO_3$, $SO_3$ can further react with $NH_3$ and $H_2O$ to form $(NH_4)_2SO_4$ and $NH_4HSO_4$ to cover the catalyst surface, resulting in the reduction in surface area, pore volume and pore size of the catalyst, and causing reversible activity loss. (ii) The $SO_2$ reacts with the active components (mainly transition metals) on the catalyst surface to form metal sulfates and induce a more severe and irreversible loss of activity. (iii) $SO_2$ showed a strong competitive adsorption effect to NO on the catalyst surface, which would reduce the formation of SCR intermediates and its catalytic efficiency, in order to improve the sulfur resistance of low-temperature SCR catalyst, which can be mainly taken in the following four ways.

### 4.1.1. Reduction in SO₂ Adsorption/Oxidation

The addition of elements, such as Co, Sn, Cu, V, Cr, Fe, as additives to the catalyst, can create a synergistic effect between the two metals, which can improve the sulfur resistance of the catalyst to some extent.

Zhang et al. [40] found that the Co-modified $Ce/MnO_2$ catalyst exhibited excellent $SO_2$ tolerance due to the addition of Co, which inhibited the sulfation of $CeO_2$ and $MnO_2$ to $CeSO_4$ and $MnSO_4$ relieved the reduction of chemisorbed oxygen, and restrained the chemisorption of $SO_2$ on the catalyst.

Hao et al. [41] prepared a Sn-doped $CeMoO_x$ catalyst that revealed better redox ability and higher Brønsted acid sites, which greatly reduced the adsorption of $SO_2$.

$MnSO_4$ formation on the $(Cu_{1.0}Mn_{2.0})_{1-\delta}O_4$ spinel was significantly alleviated by Cu doping, mainly due to the reduction in the amount of adjacent Mn. Furthermore, because of electron transfer between Cu and Mn cations within the spinel lattice ($Cu^{2+} + Mn^{3+} \rightleftarrows Mn^{4+} + Cu^{+}$), the $(Cu_{1.0}Mn_{2.0})_{1-\delta}O_4$ spinel maintained a high ratio of $Mn^{4+}/Mn_{total}$ on the surface and good low-temperature SCR activity under $SO_2$-containing conditions [42].

Jiang et al. [43] doped $V_2O_5$ into the Mn-Ce(0.4)/AC catalyst to enhance the surface acidity of the catalyst and enrich its surface chemisorbed oxygen, which accelerated the SCR reaction. The accelerated SCR reaction and stronger surface acidity inhibited the competitive adsorption of $SO_2$ and limited the reaction between adsorbed $SO_2$ and $NH_3$ species. Furthermore, vanadium oxide clusters over the catalyst partially prevented sulfation of the dispersed Mn-Ce solid solution by $SO_2$.

The $MnCrO_x$/Sepiolite catalyst has more acidic sites and oxygen vacancies, as well as a stronger redox ability compared to the pure $MnO_2$/Sepiolite catalyst. It was found that the addition of Cr could reduce the adsorption strength of $SO_2$ on the catalyst surface, thus increasing the resistance of the catalyst to $SO_2$ poisoning and reducing the activation energy [44].

In addition, Wu et al. [45,46] revealed that catalysts with a Keggin structure possessed more surface Brønsted and Lewis acid sites. These catalysts showed significantly improved performance in SCR reaction and resistance to $SO_2$ poisoning, mainly because the super acidity and Keggin structure inhibited $SO_2$ adsorption.

Wang et al. [47] found that the addition of Fe and Co to Mn-Ce/$TiO_2$ significantly reduced the adsorption of $SO_2$. The uniform distribution of Fe and Co on the surface of Mn-Ce/$TiO_2$ prevented the diffusion of $SO_2$ to the inner layer of the catalyst. In the presence of $SO_2$, the SCR reaction over 2Fe$_4$Co-MCT catalyst mainly followed the Eley–Rideal (E–R) mechanism, which the adsorbed $NH_3$ and $NH_4^+$ species reacted with gaseous NO species to produce unstable $NH_4NO_2$, followed by the rapid decomposition to $N_2$ and $H_2O$.

Inhibiting the oxidation of $SO_2$ to $SO_3$ can further avoid the formation of $NH_4HSO_4$, resulting in a reduction in the $SO_2$ poisoning over the catalyst.

Wang et al. [48] reported that doping Sm into the $MnCeTiO_x$ catalyst could increase oxygen vacancies and transfer electrons to $Mn^{4+}$ and $Ce^{4+}$, which facilitates the formation of active adsorbed $NO_2$, binary nitrate, and bridging nitrate intermediates. Meanwhile, it inhibited the oxidation of $SO_2$ by $Mn^{4+}$ and $Ce^{4+}$, leading to the suppression of $SO_2$ poisoning of the catalyst (Figure 1).

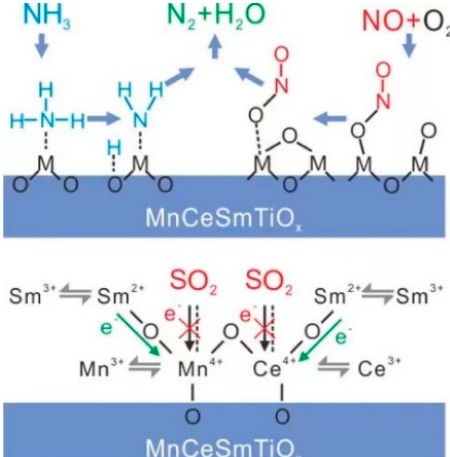

**Figure 1.** Proposed mechanism for the SCR reaction on $MnCeTiO_x$. Proposed mechanism for the suppressed oxidation of $SO_2$ to $SO_3$ on the Sm-modified $MnCeTiO_x$. These schematic diagrams are reproduced with permission from [48]. Copyright: 2020 American Chemical Society.

Hou et al. [49] prepared a La-modified $TiO_2$ carrier by the sol–gel method, and then prepared a Fe-Mn/$TiO_2$(xLa) catalyst, which significantly improved $SO_2$ resistance. It is mainly accounted for the fact that the addition of La inhibited the growth of $TiO_2$ particles, which resulted in a distinct layered structure and more mesopores of $TiO_2$, thus accelerating the decomposition of ammonium sulfate. Meanwhile, the increase in Brønsted acid sites and the electron transfer between La and Fe/Mn in the La-doped catalyst inhibited the adsorption and oxidation of $SO_2$ on the catalyst, weakened the sulfation of the active components, and thus improved $SO_2$ resistance.

Zhu et al. [50] prepared a Ce-based catalyst by the solid-state ball milling method using $Ce_2(C_2O_4)_3$ as a precursor. It was found to have a large specific surface area and pore volume. In addition, there was a large amount of $Ce^{3+}$ and surface-adsorbed oxygen, which promoted the adsorption and activation of NO. Stronger Lewis acid sites resulted in the enhanced adsorption of $NH_3$ and the reduced adsorption and oxidation of $SO_2$.

Liu et al. [51] found that, by embedding $Mn_3O_4$ nanoparticles in a defect-rich graphitic carbon framework ($Mn_3O_4$@G-A) through a MOF-derived confinement–decomposition–oxidation strategy, electron transfer from graphitic carbon to $Mn_3O_4$ through Mn–O–CGD bonds modulated the electronic properties of $Mn_3O_4$, resulting in a higher vacancy for-

mation energy and weaker oxidation ability of $Mn_3O_4$@G-A, which in turn improved $SO_2$ resistance.

Wu et al. [8] used a high shear method to uniformly mix $TiO_2$ particles with EG as a carrier and then loaded vanadium, which increased the surface area and the number of Brønsted acid sites over the catalyst and improved the surface $VO^{2+}$ as well as adsorbed oxygen species, thus reducing the adsorption and oxidation of $SO_2$.

### 4.1.2. Increasing the Adsorption of Active Intermediate Species with the Coexistence of Sulfate Species

Xu et al. [52] showed that K-modified activated carbon enhanced the adsorption of NO. Furthermore, the additional formation of nitro compounds, singlet nitrate and nitrous oxide promoted the denitrification reaction pathway. The catalytic dissociation activity of $N_2O$ on AC–K brought the formation and decomposition of $N_2O$ into a dynamic equilibrium. Thus, it provided a good resistance to sulfur dioxide.

Regarding the Ho-modified Mn/Ti catalyst, the influence of $SO_2$ on the adsorption and activation of NO on the catalyst surface was weak, and the adsorption and activation of $NH_3$ were not affected, only the reactions obeying the L–H mechanism were inhibited. Therefore, the 0.2HoMn/Ti catalyst had a better $SO_2$ resistance [53].

Yu et al. [54] found that $NH_4^+$ in $NH_4HSO_4$ deposited on the surface of the $\gamma$-$Fe_2O_3$ catalyst readily reacted with NO + $O_2$, which suppressed the continued accumulation of $NH_4HSO_4$, and an equilibrium between the deposition and consumption of $NH_4HSO_4$ on the surface of $\gamma$-$Fe_2O_3$ was established. While $SO_4^{2-}$ from ammonium sulfate continued to combine with iron ions to form iron sulfate on the catalyst surface, the formation of iron sulfate species increased the number of acid sites and enhanced the acidity intensity, thus facilitating the E–R reaction pathway. The amount of oxygen species adsorbed on the surface also increased. All of these processes are important in enhancing the efficiency of SCR.

### 4.1.3. Building Sacrificial Sites to Conserve Active Sites

The reaction of $SO_2$ and active $CeO_2$ to form $Ce_2(SO_4)_3$ is the decisive reason for $SO_2$ poisoning on the Ce-based catalyst. The addition of $Fe_2O_3$ to the Ce-$Nb_2O_5$ catalyst made it possible to act as a sacrificial victim, which reacted preferentially with $SO_2$, thus releasing the active species [55]. Kang et al. [56] also found, in this phenomenon, that the addition of $Fe_2O_3$, in the presence of $SO_2$, protected the active $CeO_2$, inhibited sulfate deposition and improved the catalytic performance of the wollastonite-supported $CeO_2$-$WO_3$. Furthermore, the addition of Fe can increase the surface acidity, surface oxygen mobility and the amount of surface adsorbed oxygen of the catalyst. Meanwhile, the $SO_4^{2-}$ formation rate can be greatly reduced, thus reducing the sulfur poisoning of the catalyst [57].

In addition, Cao et al. [58] revealed that loading niobium on CeWTi complex oxides reduces the formation of metal sulfates and protects $Ce^{x+}$ from sulfur dioxide attacks. Wang et al. [59] discovered that $CeO_2$-modified $MnCoO_x$ microflowers ($MnCoCeO_x$) with increased ratios of $Ce^{3+}/Ce^{n+}$, $Mn^{4+}/Mn^{n+}$ and $O_\alpha/(O_\alpha + O_\beta)$ prevented the sulfation of $MnCoO_x$ into metal sulfate species and improved the anti-sulfur properties of the catalyst.

Fabricated nanomaterials with a core–shell structure or a special structure can usually serve as a protective layer for the active component. Cai et al. [60] found that the $Fe_2O_3$ shell significantly inhibited the deposition of sulfate species on the surface of multi-shell $Fe_2O_3$@$MnO_x$@CNTs as compared with $MnO_x$@CNT catalyst, showing excellent $SO_2$ tolerance. Shao et al. [61] showed that H-$MnO_2$ with a hollow cavity structure could provide shuttle space for oxidation and improved mass adsorption and conversion, which can lead to better NO conversion and $SO_2$ resistance. Table 1 lists some typical $SO_2$-tolerant SCR catalysts in the literatures are summarized as following.

**Table 1.** Typical $SO_2$-tolerant catalysts reported in the literature.

| Catalysts | Reaction Condition | NO Conversion before and after Introducing $SO_2$ | Reference |
|---|---|---|---|
| $Sn/0.2\text{-}CeMoO_x$ | T = 225 °C, $[NH_3]$ = $[NO]$ = 500 ppm, $[O_2]$ = 5%, $[SO_2]$ = 200 ppm, GHSV = 100,000 $h^{-1}$ | Decline from 99% to 91% | [41] |
| $(Cu_{1.0}Mn_{2.0})_{1-\delta}O_4$ | T = 200 °C, $[NH_3]$ = $[NO]$ = 500 ppm, $[O_2]$ = 3%, $[SO_2]$ = 50 ppm, $[H_2O]$ = 5 vol%, GHSV = 100,000 $h^{-1}$ | Maintained at ~87% | [42] |
| $Mn\text{-}Ce(0.4)\text{-}V/AC$ | T = 200 °C, $[NH_3]$ = $[NO]$ = 500 ppm, $[O_2]$ = 5%, $[SO_2]$ = 100 ppm, $[H_2O]$ = 10 vol%, GHSV = 18,000 $h^{-1}$ | Maintained at ~80% | [43] |
| $MnCrO_x/Sepiolite$ | T = 220 °C, $[NH_3]$ = $[NO]$ = 1000 ppm, $[O_2]$ = 5%, $[SO_2]$ = 200 ppm, GHSV = 35,000 $h^{-1}$ | Decline from 87% to 85% | [44] |
| $V_2O_5\text{-}MoO_3/TiO_2$ | T = 250 °C, $[NH_3]$ = $[NO]$ = 1000 ppm, $[O_2]$ = 5%, $[SO_2]$ = 1000 ppm, $[H_2O]$ = 10 vol%, GHSV = 40,000 $h^{-1}$ | Decline from 100% to 83% | [45] |
| $2Fe_4Co\text{-}MCT$ | T = 200 °C, $[NH_3]$ = $[NO]$ = 500 ppm, $[O_2]$ = 6%, $[SO_2]$ = 200 ppm, $[H_2O]$ = 10 vol%, GHSV = 12,000 $h^{-1}$ | Decline from 98% to 90% | [47] |
| $MnCeSmTiO_x$ | T = 200 °C, $[NH_3]$ = $[NO]$ = 500 ppm, $[O_2]$ = 5%, $[SO_2]$ = 200 ppm, $[H_2O]$ = 5 vol%, GHSV = 80,000 $h^{-1}$ | Maintained at ~70% | [48] |
| $Fe–Mn/TiO_2(0.02La)$ | T = 200 °C, $[NH_3]$ = $[NO]$ = 1000 ppm, $[O_2]$ = 7%, $[SO_2]$ = 100 ppm, GHSV = 30,000 $h^{-1}$ | Maintained at ~99% | [49] |
| $CeTiO_x$ | T = 300 °C, $[NH_3]$ = $[NO]$ = 1000 ppm, $[O_2]$ = 6%, $[SO_2]$ = 175 ppm, $[H_2O]$ = 6 vol%, GHSV = 30,000 $h^{-1}$ | Decline from 100% to 70% | [50] |
| $Mn_3O_4@G\text{-}A$ | T = 160 °C, $[NH_3]$ = 660 ppm, $[NO]$ = 600 ppm, $[O_2]$ = 5%, $[SO_2]$ = 50 ppm, GHSV = 20,000 $h^{-1}$ | Decline from 100% to 70% | [51] |
| $0.2HoMn/Ti$ | T = 180 °C, $[NH_3]$ = $[NO]$ = 500 ppm, $[O_2]$ = 6%, $[SO_2]$ = 100 ppm, $[H_2O]$ = 10 vol%, GHSV = 20,000 $h^{-1}$ | Decline from 100% to 80% | [53] |
| $CeO_2\text{-}Fe_2O_3\text{-}Nb_2O_5$ | T = 300 °C, $[NH_3]$ = $[NO]$ = 667 ppm, $[O_2]$ = 5%, $[SO_2]$ = 200 ppm, GHSV = 120,000 $mL \cdot g^{-1} \cdot h^{-1}$ | Maintained at ~92% | [55] |
| $Fe(4)@CeW/H$ | T = 300 °C, $[NH_3]$ = $[NO]$ = 500 ppm, $[O_2]$ = 5%, $[SO_2]$ = 100 ppm, $[H_2O]$ = 8 vol%, GHSV = 40,000 $h^{-1}$ | Decline from 100% to 82% | [56] |
| $Mn_1Fe_{0.25}Al_{0.75}Ox$ | T = 150 °C, $[NH_3]$ = $[NO]$ = 500 ppm, $[O_2]$ = 5%, $[SO_2]$ = 50 ppm, GHSV = 60,000 $h^{-1}$ | Maintained at ~80% | [57] |
| $NbCeWTi$ | T = 270 °C, $[NH_3]$ = $[NO]$ = 500 ppm, $[O_2]$ = 5%, $[SO_2]$ = 200 ppm, $[H_2O]$ = 5 vol%, GHSV = 80,000 $h^{-1}$ | Decline from 92% to 71% | [58] |
| $Fe@Mn@CNTs$ | T = 240 °C, $[NH_3]$ = $[NO]$ = 550 ppm, $[O_2]$ = 5%, $[SO_2]$ = 100 ppm, $[H_2O]$ = 10 vol%, GHSV = 20,000 $h^{-1}$ | Decline from 100% to 91% | [60] |

### 4.1.4. Promoting the Decomposition of Sulfates

Some additives can facilitate the decomposition of surface $(NH_4)_2SO_4$ and $NH_4HSO_4$, leading to the improvement of the sulfur resistance of the catalyst. Fan et al. [62] found that $Al_2O_3$ added into $MnO_x$ not only inhibited the reaction between $MnO_x$ and $SO_2$ and promoted the decomposition of $NH_4HSO_4$, but also led, to a certain degree, to a decrease in the thermal stability of the adsorbed $SO_2$ species. The $CeO_2$ or $WO_3$ incorporated into a $V_2O_5/TiO_2$ catalyst could promote the decomposition of the $NH_4HSO_4$ deposited on the catalyst surface [63,64]. Additionally, Ye et al. [65] revealed that $Nb_2O_5$ and $Sb_2O_5$ doped into the $V_2O_5\text{-}WO_3/TiO_2$ catalyst could accelerate the reaction of the deposited $NH_4HSO_4$ with gaseous NO over the catalyst.

Furthermore, Guo et al. [66] also found that the larger pore size of the carrier could greatly ease the deposition of ABS. Using mesoporous silica SBA-15 as the support material, the decomposition of ABS was significantly accelerated with the enlarged pore size, largely because the larger pore size generated higher vapor pressure, which facilitated the vaporization and decomposition of ABS (Figure 2).

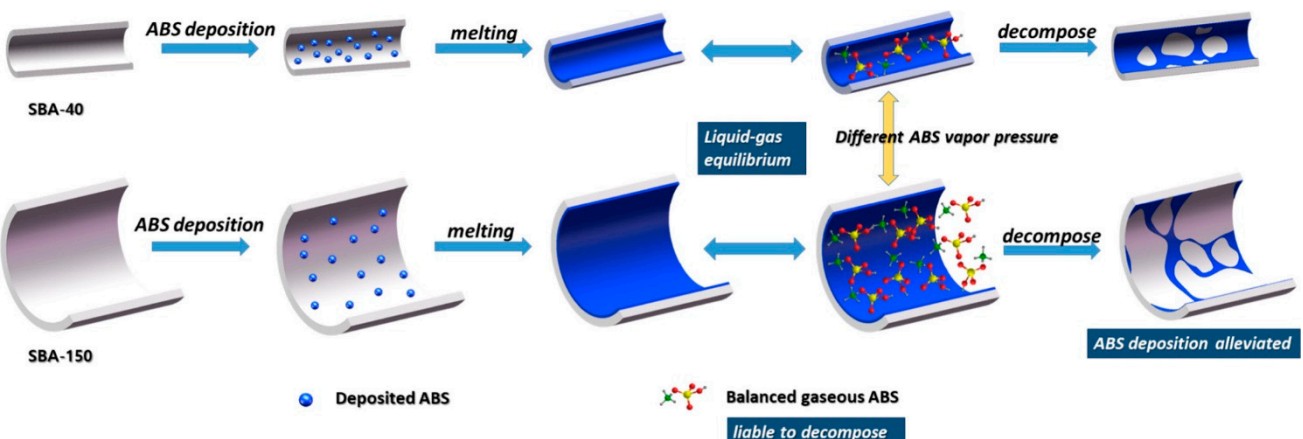

**Figure 2.** Illustration of ABS decomposition behaviors on SBA-15 with different pore sizes. These schematic diagrams are reproduced with permission from [66]. Copyright: 2019 American Chemical Society.

### 4.2. $H_2O$ Poisoning

The effect of $H_2O$ on SCR activity is mainly due to the following three reasons. (i) The competitive adsorption of $H_2O$ and NO or $NH_3$ decreased the $NH_3$-SCR reaction activity. (ii) $H_2O$ could promote the formation of hydroxyl groups (OH) on the catalyst surface, thus reducing the number of active sites and inhibiting the adsorption of $NH_3$ and NO species and the reaction between them [67]. (iii) $H_2O$ could accelerate the formation of $NH_4HSO_4$ when both $SO_2$ and $H_2O$ are present simultaneously. There are several ways to reduce the adsorption of $H_2O$ on the SCR catalyst surface and the active center to improve the $H_2O$ resistance of the catalyst [68].

#### 4.2.1. Doping with Rare Earth Metal Elements or Transition Metal Elements

Zhang et al. [69] gave an example to improve the $H_2O$ tolerance in Mn-based materials. The Ce-$MnO_2$ catalyst has good resistance to $H_2O$, and both Ce-containing catalyst has a more than 90% $NO_x$ conversion below 150 °C in the presence of superior $H_2O$. The competitive adsorption of $NO_x$ on the surface of Ce-$MnO_2$ was stronger than that of $H_2O$. However, $NH_3$ and $H_2O$ did not have any competitive adsorption on both. Ma et al. [70] reported that doping Ce on the $Cu_{0.02}Fe_{0.2}TiO_x$ catalyst increased the $H_2O$ resistance of the catalyst dramatically, mainly because the amount of Lewis acid increased largely and the interaction between $H_2O$ and Lewis acid sites was weaker than that of Brønsted acid sites.

Guo et al. [71] added Fe to the Mn-Eu catalyst, which showed excellent resistance to $H_2O$, even at a high gas hourly space velocity (GHSV) of 75,000 $h^{-1}$. The NO conversion remained at 90% at 230 °C in the presence of 15% $H_2O$ after 50 h, and the detrimental effect of $H_2O$ could be quickly eliminated after its removal.

#### 4.2.2. Doping or Modification of the Catalyst Surface with Hydrophobic Materials or Moieties

Zhang et al. [72] showed a way to enhance the $H_2O$ resistance with the assistance of hydrophobic polytetrafluorethylene (PTFE) materials. The Mn-PTFE composite catalyst exhibited a good tolerance to $H_2O$, which was related to the hydrophobic surface generated, making the catalyst surface active sites less likely to be consumed by $H_2O$. Additionally, Wang et al. [73] found that polyethylene oxide-polypropylene-polyethylene oxide (P123)-modified Mn-MOF-74 demonstrated excellent NO conversion of up to 92.1% in the presence of 5% $H_2O$ at 250 °C. The undecorated Mn-MOF-74, under the same conditions, obtained only 52% NO conversion. Alternatively, the introduction of a hydrophobic group (-$CH_3$) on the ligand of Mn-MOF-74 could similarly reduce the effect of $H_2O$.

### 4.2.3. Regulating the Crystal Phase and Structure

Hu et al. [74] found that $\alpha$-MnO$_2$ has a better resistance to H$_2$O than $\beta$-MnO$_2$. Because $\alpha$-MnO$_2$ has a special semi-tunnel structure and exposed active (110) surface, a double ionic cluster can be easily formed over the surface of $\alpha$-MnO$_2$, showing good hydrophobicity in the presence of H$_2$O. Qiu et al. [75] showed that 3D-MnCo$_2$O$_4$ has a good resistance to H$_2$O and about 96% NO conversion can be achieved over the catalyst at 200 °C in the presence of 10% H$_2$O. Furthermore, Tang et al. [76] revealed that the resistance of H$_2$O over MnO$_x$ was also related to the synthesis method. MnO$_x$ prepared by the Co-precipitation method has a better H$_2$O resistance than MnO$_x$ prepared by the solid phase reaction method at 80 °C (10% H$_2$O, 47,000 h$^{-1}$) because the former has a larger specific surface area.

### 4.3. Alkali Poisoning

An alkali metal has a toxic effect on the catalyst mainly because it occupied the acidic sites of the catalyst, reducing the number of acidic sites and decreasing the adsorption of NH$_3$ [77,78]. The following two approaches are available to reduce the toxicity of alkali metals to catalysts.

### 4.3.1. Providing More Acidic Sites

The deposition of an acidic promoter on the catalyst surface can increase the number of acidic sites on the surface as well as binding to toxic metals, leaving the active metal sites free, thus improving the resistance of SCR catalysts to alkali metals [79]. Kang et al. [80] indicated that CuO has an excellent reducing property and surface acidity and the CuO modification on CeTiO$_x$ catalyst effectively improved its low-temperature catalytic activity and resistance to alkali metal poisoning. Co modification on 0.3K3Mn10Fe/Ni catalyst resulted in the increase in the ratio of NiO/Ni, the concentration of Fe$^{3+}$ and Mn$^{4+}$, lattice oxygen percentage, unsaturated Ni atoms content, and the number of Ni-Mn/Fe composite oxides structure defects, which caused an enhancement of the synergistic effect between Ni and Mn/Fe. The increase in Brønsted acid sites results in the enhancement of the mid-low temperature redox ability of the catalyst, thus significantly improving the K poisoning resistance of the 3Mn10Fe/Ni catalyst. For the 3Mn10Fe/Ni catalyst, as the addition of Co is 0.2%, the K poisoning resistance of the catalyst is optimal, and the de-NO$_x$ activity reaches 98% at 220 °C [81]. Nie et al. [82] discovered that V$_2$O$_5$-Ce(SO$_4$)$_2$/TiO$_2$ performs better in a SCR reaction than V$_2$O$_5$-WO$_3$/TiO$_2$ when KCl is deposited on the catalyst surface. This is because the V$_2$O$_5$-Ce(SO$_4$)$_2$/TiO$_2$ catalyst has a stronger surface acidity. Moreover, Wang et al. [83] clearly identified that the acidic H-SAPO-34 with abundant Brønsted acid sites ensured a high dispersion of active Cu$^{2+}$ sites, while providing a large number of ion-exchange protons to replace the K, Ca, Pb ions in the zeolite framework. The Ce species promoted the electron-transfer effect with Cu active sites and improved redox cycling. It could also bind to Ca and Pb toxicants and promote the resistance of the catalysts to harmful elements (Figure 3). Accordingly, the dual promotion of H-SAPO-34 and Ce enabled the CuCe/H-SAPO-34 catalyst to effectively control NO$_x$ emissions in the presence of alkali metals.

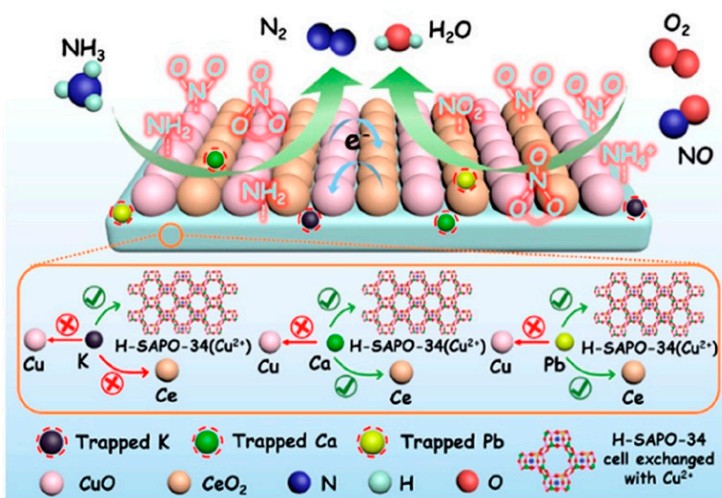

**Figure 3.** Proposed mechanism of NO$_x$ reduction in the presence of alkaline and heavy metals over CuCe/H-SAPO-34 catalysts. These schematic diagrams are reproduced with permission from [83]. Copyright: 2020 American Chemical Society.

### 4.3.2. Building Alkaline Capture Sites to Protect Active Sites

Zha et al. [84] reported that the Fe-OMS-2 catalyst was found to have a better alkaline tolerance due to having more redox species, more acidic sites and a stronger adsorption of NO$_x$ species as compared with the undoped MnO$_x$ octahedral molecular sieve (OMS-2) catalyst. Meanwhile, K$^+$ ions can be trapped by cryptic tunneling through an ion exchange mechanism, resulting in a more stable system structure. Wu et al. [85] synthesized an alkali metal-resistant deNO$_x$ catalyst that utilized ion-exchanged titanate nanotubes as supports to stabilize alkali metal ions in the interlayer of titanate nanotubes. Yan et al. [86] proved that the OMS-5(H)@TiO$_2$ catalyst has a good resistance to alkali metals. The special structure of the MnO$_x$ octahedral molecular sieve (OMS-5(H)) could trap alkali metals and the acid treatment increased the acid sites, which enabled the catalyst to resist alkali metals in the form of ion exchange. The decoration of TiO$_2$ further increased the strength of Lewis acid sites, allowing more active intermediate species to participate effectively in the NO$_x$ reaction.

The CuNbTi complex oxide catalyst exhibited a good alkali resistance. The K atom preferentially interacted with Nb–OH and Nb=O, which protected the active species. K$_2$O can be trapped on TiO$_2$ supported by the addition of Niobium species, while isolating Cu$^{2+}$, as the active sites play a key role in yielding the remaining activity [87].

The works discussed above have laid the foundation for the development of superior low-temperature SCR catalysts, and much progress has been made in catalyst resistance to poisoning. However, successful applications in flue gas containing one or more components of SO$_2$, H$_2$O, and alkali metals have rarely been reported. The tolerance of SO$_2$ and H$_2$O is closely related to the reaction temperature, and most experiments have been performed above 200 °C, which may not have practical applications. At least 10% or even 30% of H$_2$O is actually present in flue gas. The experimental H$_2$O content and stability testing time are often insufficient. Researchers have studied the effect of alkali on SCR catalysts mainly by impregnating the catalyst with alkali metal precursors followed by a calcination treatment. The deactivation process of alkali in fly ash in actual flue gas may be different from the alkali poisoning process of the impregnation method. Therefore, bringing the anti-poisoning catalysts closer to practical applications is a challenge and more efforts are needed in this area.

## 5. Industrial Applications of Low-Temperature SCR Catalysts in China

In the last decade, the research and development of low-temperature SCR catalysts and deNO$_x$ engineering technology have achieved great progress and success in China. Commercial low-temperature SCR catalyst products are widely used in deNO$_x$ project in

the coking, sintering, waste incineration and lime kiln industries. Table 2 lists the low-temperature SCR catalysts produced by major domestic enterprises and their engineering applications, including demonstration projects.

**Table 2.** Low-temperature SCR catalyst research units and commissioning (test) information [88].

| Catalyst | Company | Cooperative Unit | Temperature Window in Which It Is Run | Application Examples |
|---|---|---|---|---|
| "Fangxin" Vanadium-Titanium Catalyst | Advanced E-catal. Co., Ltd. | Beijing University of Technology | 160–400 °C | Zhanjiang Baosteel Group Co., Ltd. 7 M top mounted coking furnace-DeNO$_x$ |
| Low-temperature honeycomb and foam type non-vanadium catalysts | Zhongneng Guoxin (Beijing) Technology Development Co. | Tsinghua University | 150–400 °C | Cement kiln low-temperature SCR denitrification |
| Mn/FA-PG non-vanadium catalyst | Hefei Chenxi Environmental Protection Engineering Co. | Hefei University of Technology | 180–300 °C | Shandong Tiexiong Xinsha Energy Co. Coke oven flue gas |
| MnO$_x$-CoO$_x$(CeO$_x$)TiO$_2$ Honeycomb/stacked bar catalysts | Shanghai Han Yu Environmental Materials Co. | Zhejiang University | 130–260 °C | Glass kiln and petrochemical field |

The first successful application of low-temperature SCR is the 7M top mounted with 4 × 65 hole coking furnace deNO$_x$ project located in Zhanjiang Baosteel Group Co., Ltd., which also is the world's first industrialized deNO$_x$ project using a low-temperature SCR catalyst with a running temperature of 180–200 °C for the purification of the coking flue gas [89].

The project was designed to produce about 3.4 million tons of dry coke per year at the largest scale, using 4 × 65-hole JNX2-70-2 type reheat top-loading coke ovens. It required to simultaneously control SO$_2$, NO$_x$ and PM emissions in flue gas to meet strict emission regulations. As shown in Figure 4, the flue gas flows through the SDA (Spray Drying Absorption) desulfurization system and the integrated reactor of the dust filter close-coupled with the SCR catalyst, and then finally returns to the chimney. The SCR reactor is designed with 16 independent reaction chambers, each of which can be alternately individually heated to 350 °C to decompose the NH$_4$HSO$_4$ generated in the reaction, realizing the in situ thermal regeneration of the SCR catalyst and ensuring the normal operation of the low-temperature SCR catalyst.

The engineering purification equipment of this project was formally put into operation in November 2015, and to date, the flue gas SO$_2$ emission concentration decreased from 80–150 mg/Nm$^3$ to 20 mg/Nm$^3$, the total SO$_2$ emission decreased from 180 kg/h to 24 kg/h, the desulfurization efficiency reached 90%, cutting SO$_2$ by about 30,000 tons/year. The NO$_x$ emission concentration decreased from ca. 500 mg/Nm$^3$ to ca. 100 mg/Nm$^3$, the total NO$_x$ emission decreased from 600 kg/h to 120 kg/h with deNO$_x$ efficiency of ca. 80 %, and the NO$_x$ emission was reduced by about 100,000 tons/year, resulting in significant environmental, social and economic benefits. The catalyst used in this project is a V–Ti-based low-temperature SCR catalyst developed by He's group at Beijing University of Technology. This catalyst was officially produced and put into the market in 2012 by Advanced E-catal. Co. Ltd., located in Beijing, China. In addition to the application of coking flue gas denitration, the low-temperature SCR catalyst was also used in the purification of sintering flue gas. Due to the large volume of flue gas, complex pollutants, large working condition fluctuates, low flue gas temperature and high sulfur content, the denitrification of sintering flue gas faces huge technical challenges. Renfon Steel Co., Ltd., located in Tangshan, employed the engineering process of "semi-dry desulfurization of flue

gas + GGH (Gas Gas Heater) heat exchange + flue gas warming + low temperature SCR denitrification + GGH heat exchange + induced draft fan + stack emission" to control the emission of sintering flus gas. The project with low-temperature SCR denitrification was successfully put into operation to meet the strict emission regulations on 2 November 2018, using the "Fangxin" low-temperature catalyst produced by Advanced E-catal. Co. Ltd. This is the first application of a low-temperature SCR denitrification catalyst on a sintering flue gas denitrification project in China [90]. The inlet $NO_x$ concentration of the SCR reactor is ca. 200–300 mg/Nm$^3$ and the outlet $NO_x$ concentration is stabilized below 20 mg/Nm$^3$ with ammonia escape smaller than 1 ppm, which can realize the ultra-low emission of sinter flue gas $NO_x$ or even the near-zero emission requirement in the future. The project also showed that the low-temperature SCR catalyst can eliminate the dioxin emissions in its application. The project was a huge success as all pollutant emissions are lower than the values stipulated by the national environmental regulations, meeting the expected targets.

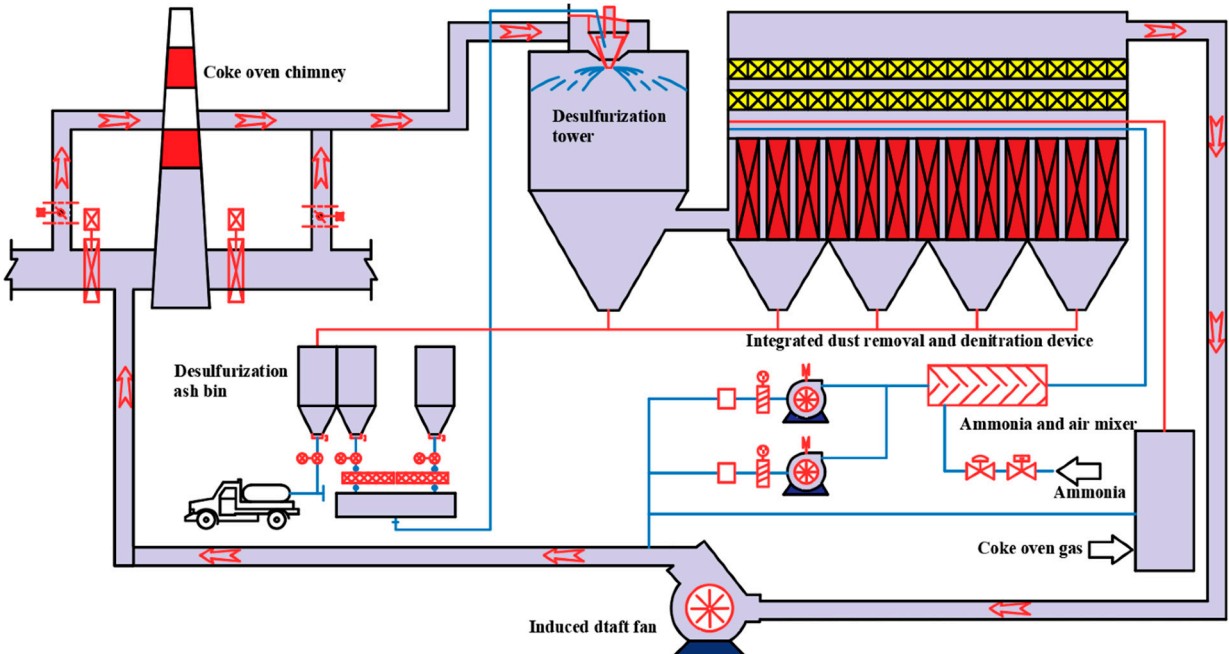

**Figure 4.** Coking deNO$_x$ project flow chart using the low-temperature SCR catalyst in Zhanjiang Baosteel Group Co.

## 6. Concluding Remarks and Future Perspectives

Low-temperature SCR technology has been proven as an effective and crucial way to abate the $NO_x$ emitted from non-electric industries, based on its several successful applications in China in the past years. There are still many problems and challenges in gaseous pollutant control in non-power industries, which need to be studied further.

Moreover, new technology with a lower energy consumption and muti-functions needs to be developed to update the demands of deep treatments of $NO_x$. The following issues need to be addressed in future research:

(1)　The SO$_2$ poisoning of the low-temperature SCR catalyst is still a vast problem in practice, especially in flue gas with high humidity.

(2)　NH$_3$ slips from the deNO$_x$ system, releasing ammonium salts to the environment. The selective oxidation of the NH$_3$ catalyst should be developed, which should be coupled with the SCR catalyst in practical applications.

(3)　In some industries, there may also be harmful elements, such as As, Hg, and Pd, in flue gas, which will affect the service life of the SCR catalysts, but also cause air pollution. Therefore, in the design of the catalyst and pollution control engineering,

new technologies should be developed to solve the complex smoke pollution problems with, for example, the development of multi-functional SCR catalysts.

(4)　In flue gas, such as sintering, in addition to $NO_x$, there is a relatively high concentration of CO. How to use the oxidation of CO to increase the temperature of the SCR catalyst bed or how to use CO as a reducing agent to treat $NO_x$ are also very important topics that need to be studied in the future.

**Author Contributions:** Literature search, H.Z. and K.L.; Article structure design, H.Z., L.S., W.Q. and H.H.; Manuscript writing, H.Z. and L.S.; Graphics production, H.Z.; Creating diagrams, R.W.; Manuscript revision, H.H. All authors have read and agreed to the published version of the manuscript.

**Funding:** This research received no external funding.

**Data Availability Statement:** Not applicable.

**Conflicts of Interest:** The authors declare no conflict of interest.

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
