# Peer review of "Low-Temperature SCR Catalyst Development and Industrial Applications in China"

_catalysts, doi:10.3390/catal12030341_

Round 1
Reviewer 1 Report
The review submitted to the Catalysts journal is devoted to industrial application of selective catalytic reduction (SCR) technology in China. SCR of NOx by ammonia over V2O5-WO3(MoO3)/TiO2 catalysts is currently the most developed and widely applied technology for NOx abatement. However, there are still some issues, which need solving, such as insufficient NOx removal at temperatures below 300°C, sulphur and alkali metal poisoning, low hydrothermal stability, NH3-slip and etc. The topic is of interest to scientific community and is perfectly in line with the Catalysts journal; therefore I support the potential publication of this review. However, there are some major points, which need revising before acceptance: 1) The title should be revised. The use of phrase ‘SCR catalysts technology development’ seems to be not appropriate (here and further). I think that it would be better something like “Low temperature SCR catalyst development and industrial application in China” or “Low temperature SCR technology development and industrial application in China”. 2) It would be very helpful, if the authors give a table, in which the information about activity/ stability and operating temperature range of the different SCR catalysts is summarized. The information should be analyzed critically. 3) In my opinion, it would be better to combine the following two sections: “Conclusions” and “Challenges and future prospects”. 4) References in text should be numbered using Arabic numbers. 5) In the current state, there are some typographical errors; therefore, the authors are advised to revise the whole manuscript carefully.
Author Response
Ms. Ref. No.: 1619597
Response Letter
Dear Editor,
Thank you and the Reviewers for the constructive comments. After considering your and their remarks carefully, we have properly revised our manuscript. The changes that we made in the revised manuscript are highlighted in RED for easy reference. Our responses are as follows:
Referee #1: The review submitted to the Catalysts journal is devoted to industrial application of selective catalytic reduction (SCR) technology in China. SCR of NOx by ammonia over V2O5-WO3(MoO3)/TiO2 catalysts is currently the most developed and widely applied technology for NOx abatement. However, there are still some issues, which need solving, such as insufficient NOx removal at temperatures below 300°C, sulphur and alkali metal poisoning, low hydrothermal stability, NH3-slip and etc. The topic is of interest to scientific community and is perfectly in line with the Catalysts journal; therefore I support the potential publication of this review. However, there are some major points, which need revising before acceptance:
Comment 1: The title should be revised. The use of phrase ‘SCR catalysts technology development’ seems to be not appropriate (here and further). I think that it would be better something like “Low temperature SCR catalyst development and industrial application in China” or “Low temperature SCR technology development and industrial application in China”.
Response: Accepting the reviewers' comment, we have revised the title of the article.
Modification: Page 1 Line 2~3:
Low temperature SCR catalysts development and industrial application in China
Comment 2: It would be very helpful, if the authors give a table, in which the information about activity/ stability and operating temperature range of the different SCR catalysts is summarized. The information should be analyzed critically.
Response: Accepting the reviewers' comment, we have produced a table summarising the reaction conditions and activity of the different SCR catalysts.
Modification: Page 5~6 Line 226:
Table 1. Typical SO2-tolerant catalysts reported in the literatures
catalysts |
reaction condition |
NO conversion before and after introducing SO2 |
ref |
Sn/0.2-CeMoOx |
T = 225 °C, [NH3] = [NO] = 500 ppm, [O2] = 5%, [SO2] = 200 ppm, GHSV = 100,000 h-1 |
decline from 99% to 91% |
41 |
(Cu1.0Mn2.0)1−δO4 |
T = 200 °C, [NH3] = [NO] = 500 ppm, [O2] = 3%, [SO2] = 50 ppm, [H2O] = 5 vol%, GHSV = 100,000 h-1 |
maintaining at ~87% |
42 |
Mn-Ce(0.4)-V/AC |
T = 200 °C, [NH3] = [NO] = 500 ppm, [O2] = 5%, [SO2] = 100 ppm, [H2O] = 10 vol%, GHSV = 18,000 h-1 |
maintaining at ~80% |
43 |
MnCrOx/Sepiolite |
T = 220 °C, [NH3] = [NO] = 1000 ppm, [O2] = 5%, [SO2] = 200 ppm, GHSV = 35,000 h-1 |
decline from 87% to 85% |
44 |
V2O5-MoO3/TiO2 |
T = 250 °C, [NH3] = [NO] = 1000 ppm, [O2] = 5%, [SO2] = 1000 ppm, [H2O] = 10 vol%, GHSV = 40,000 h-1 |
decline from 100% to 83% |
45 |
2Fe4Co-MCT |
T = 200 °C, [NH3] = [NO] = 500 ppm, [O2] = 6%, [SO2] = 200 ppm, [H2O] = 10 vol%, GHSV = 12,000 h-1 |
decline from 98% to 90% |
47 |
MnCeSmTiOx |
T = 200 °C, [NH3] = [NO] = 500 ppm, [O2] = 5%, [SO2] = 200 ppm, [H2O] = 5 vol%, GHSV = 80,000 h-1 |
maintaining at ~70% |
48 |
Fe–Mn/TiO2(0.02La) |
T = 200 °C, [NH3] = [NO] = 1000 ppm, [O2] = 7%, [SO2] = 100 ppm, GHSV = 30,000 h-1 |
maintaining at ~99% |
49 |
CeTiOx |
T = 300 °C, [NH3] = [NO] = 1000 ppm, [O2] = 6%, [SO2] = 175 ppm, [H2O] = 6 vol%, GHSV = 30,000 h-1 |
decline from 100% to 70% |
50 |
Mn3O4@G-A |
T = 160 °C, [NH3] = 660 ppm, [NO] = 600 ppm, [O2] = 5%, [SO2] = 50 ppm, GHSV = 20,000 h-1 |
decline from 100% to 70% |
51 |
0.2HoMn/Ti |
T = 180 °C, [NH3] = [NO] = 500 ppm, [O2] = 6%, [SO2] = 100 ppm, [H2O] = 10 vol%, GHSV = 20,000 h-1 |
decline from 100% to 80% |
53 |
CeO2-Fe2O3-Nb2O5 |
T = 300 °C, [NH3] = [NO] = 667 ppm, [O2] = 5%, [SO2] = 200 ppm, GHSV = 120,000 mL–g-1–h-1 |
maintaining at ~92% |
55 |
Fe(4)@CeW/H |
T = 300 °C, [NH3] = [NO] = 500 ppm, [O2] = 5%, [SO2] = 100 ppm, [H2O] = 8 vol%, GHSV = 40,000 h-1 |
decline from 100% to 82% |
56 |
Mn1Fe0.25Al0.75Ox |
T = 150 °C, [NH3] = [NO] = 500 ppm, [O2] = 5%, [SO2] = 50 ppm, GHSV = 60,000 h-1 |
maintaining at ~80% |
57 |
NbCeWTi |
T = 270 °C, [NH3] = [NO] = 500 ppm, [O2] = 5%, [SO2] = 200 ppm, [H2O] = 5 vol%, GHSV = 80,000 h-1 |
decline from 92% to 71% |
58 |
Fe@Mn@CNTs |
T = 240 °C, [NH3] = [NO] = 550 ppm, [O2] = 5%, [SO2] = 100 ppm, [H2O] = 10 vol%, GHSV = 20,000 h-1 |
decline from 100% to 91% |
60 |
Comment 3: In my opinion, it would be better to combine the following two sections: “Conclusions” and “Challenges and future prospects”.
Response: Accepting the reviewers' comment, we have combined and revised the sections "Conclusions" and "Challenges and future prospects".
Modification: Page 14 Line 526~546:
- Concluding remarks and future perspectives
The low temperature SCR technology has been proven as an effective and crucial way to abating the NOx emitted from the non-electric industries based on the successful applications in China for the past years. There are still many problems and challenges in the gaseous pollutant control in the non-power industry, which need to be studied further.
Moreover, developed new technology with lower energy consumption and muti-func tions is in need as the update demand of deep treatment in NOx.
(1) SO2 poisoning of the low temperature SCR catalyst is still a big problem in practice, especially in flue gas with high humidity.
(2) NH3 slip from the deNOx system gives ammonium salts in the environment. The selective oxidation of NH3 catalyst should be developed, which should be coupled with the SCR catalyst in application.
(3) In some industries, there may also be harmful elements such as As, Hg, Pd and others in flue gas, which will affect the service life of SCR catalysts, but also cause air pollution. Therefore, in the design of catalyst and pollution control engineering, new technologies should be developed to solve complex smoke pollution problems, for example, the developing multi-functional SCR catalysts.
(4) In the flue gas such as sintering, in addition to NOx, there is a relatively high concentration of CO. How to use the oxidation of CO to increase the temperature of the SCR catalyst bed, or use CO as a reducing agent to treat NOx are also very important topics needed to study in future.
Comment 4: References in text should be numbered using Arabic numbers.
Response: Accepting the reviewers' comment, we have amended all reference numbers in the text to Arabic numerals.
Modification: Please see article for reference numbers, marked in red.
Comment 5: In the current state, there are some typographical errors; therefore, the authors are advised to revise the whole manuscript carefully.
Response: Accepting the reviewers' comment, we have read the article carefully and revised some of the words and phrases.
Modification: Please see the words and phrases marked in red in the article.
We sincerely hope that the revised manuscript is now acceptable for publication in the "Catalysts".
With regards,
Yours sincerely,
Prof. Liyun Song
Department of Environmental Science,
School of Environmental and Chemical Engineering,
Faculty of Environment and Life,
Beijing University of Technology,
Beijing 100124, China.
E-mail address: songly@bjut.edu.cn (L.Y. Song)
Prof. Hong He
Laboratory of Catalysis Chemistry and Nanoscience,
Department of Environmental Chemical Engineering,
School of Environmental and Chemical Engineering,
Faculty of Environment and Life,
Beijing University of Technology,
Beijing 100124, China
E-mail: hehong@bjut.edu.cn (H. He)
Mar. 3, 2022.

Reviewer 2 Report
The authors elaborated a review about the low-temperature SCR denitrification technology in China. SCR is an important process in the control of NOx emission. The paper has a particular focus on the mechanism of catalyst poisoning in industrial applications, and the recent achievements in denitrification in the non-power Chinese industries. The review is clear, concise, and reports interesting aspects from the literature data. I think it is an interesting contribution to the state of the art of industrial NOx abatement, and I recommend the publication after minor changes and English revision.
1) the images published with the permission of other authors are not very clear, the texts in the figures are in many points not legible and clear. If you cannot improve their resolution quality, you need to draw them as new ones:
Fig 1: the two graphs at the bottom are not legible
Fig. 2: molecules and atoms in the coloured circle are non-legible at all in printed paper
Fig. 3: low resolution, difficulty to read
Fig. 4: low resolution, text in the blue rectangles nor readable
Fig. 5: too low resolution
Fig. 6 low resolution
Fig. 7 texts with low resolution
2) the conclusions appear rather incomplete, and I strongly suggest improving and expanding them, describing more clearly the results that emerged in this summary and the problems that remain and the aspects that still need to be improved pin the opinion of the authors.
2) English should be revised: only some example: lines 48 (of), 185 (compare) , 220 (react) , 222 (compete), 245 (clastering), 272-273 (ing form), 376 (adsorption ON), 379 (were)…. Etc.
incomplete sentences , lines 213-214
Author Response
Ms. Ref. No.: 1619597
Response Letter
Dear Editor,
Thank you and the Reviewers for the constructive comments. After considering your and their remarks carefully, we have properly revised our manuscript. The changes that we made in the revised manuscript are highlighted in RED for easy reference. Our responses are as follows:
Referee #2: The authors elaborated a review about the low-temperature SCR denitrification technology in China. SCR is an important process in the control of NOx emission. The paper has a particular focus on the mechanism of catalyst poisoning in industrial applications, and the recent achievements in denitrification in the non-power Chinese industries. The review is clear, concise, and reports interesting aspects from the literature data. I think it is an interesting contribution to the state of the art of industrial NOx abatement, and I recommend the publication after minor changes and English revision.
Comment 1: The images published with the permission of other authors are not very clear, the texts in the figures are in many points not legible and clear. If you cannot improve their resolution quality, you need to draw them as new ones:
Fig 1: The two graphs at the bottom are not legible;
Fig. 2: Molecules and atoms in the coloured circle are non-legible at all in printed paper;
Fig. 3: Low resolution, difficulty to read;
Fig. 4: Low resolution, text in the blue rectangles nor readable;
Fig. 5: Too low resolution;
Fig. 6: Low resolution;
Fig. 7: Texts with low resolution.
Response: Accepting the reviewers' comment, we have modified all the figures in the article to high resolution.
Modification: Please see the figures in the article.
Comment 2: The conclusions appear rather incomplete, and I strongly suggest improving and expanding them, describing more clearly the results that emerged in this summary and the problems that remain and the aspects that still need to be improved pin the opinion of the authors.
Response: Accepting the reviewers' comment, we have revised the conclusion section to summarise and suggest problems and areas for improvement.
Modification: Page 14 Line 526~546:
- Concluding remarks and future perspectives
The low temperature SCR technology has been proven as an effective and crucial way to abating the NOx emitted from the non-electric industries based on the successful applications in China for the past years. There are still many problems and challenges in the gaseous pollutant control in the non-power industry, which need to be studied further.
Moreover, developed new technology with lower energy consumption and muti-func tions is in need as the update demand of deep treatment in NOx.
(1) SO2 poisoning of the low temperature SCR catalyst is still a big problem in practice, especially in flue gas with high humidity.
(2) NH3 slip from the deNOx system gives ammonium salts in the environment. The selective oxidation of NH3 catalyst should be developed, which should be coupled with the SCR catalyst in application.
(3) In some industries, there may also be harmful elements such as As, Hg, Pd and others in flue gas, which will affect the service life of SCR catalysts, but also cause air pollution. Therefore, in the design of catalyst and pollution control engineering, new technologies should be developed to solve complex smoke pollution problems, for example, the developing multi-functional SCR catalysts.
(4) In the flue gas such as sintering, in addition to NOx, there is a relatively high concentration of CO. How to use the oxidation of CO to increase the temperature of the SCR catalyst bed, or use CO as a reducing agent to treat NOx are also very important topics needed to study in future.
Comment 3: English should be revised: only some example: lines 48 (of), 185 (compare), 220 (react), 222 (compete), 245 (clastering), 272-273 (ing form), 376 (adsorption ON), 379 (were)…. Etc. Incomplete sentences , lines 213-214.
Response: Accepting the reviewers' comment, we have read the article carefully and revised some of the words and phrases.
Modification: Please see the words and phrases marked in red in the article.
We sincerely hope that the revised manuscript is now acceptable for publication in the "Catalysts".
With regards,
Yours sincerely,
Prof. Liyun Song
Department of Environmental Science,
School of Environmental and Chemical Engineering,
Faculty of Environment and Life,
Beijing University of Technology,
Beijing 100124, China.
E-mail address: songly@bjut.edu.cn (L.Y. Song)
Prof. Hong He
Laboratory of Catalysis Chemistry and Nanoscience,
Department of Environmental Chemical Engineering,
School of Environmental and Chemical Engineering,
Faculty of Environment and Life,
Beijing University of Technology,
Beijing 100124, China
E-mail: hehong@bjut.edu.cn (H. He)
Mar. 3, 2022.